# Cinnamaldehyde-Mediated Suppression of MMP-13, COX-2, and IL-6 Through MAPK and NF-κB Signaling Inhibition in Chondrocytes and Synoviocytes Under Inflammatory Conditions

**DOI:** 10.3390/ijms252312914

**Published:** 2024-11-30

**Authors:** Jaishree Sankaranarayanan, Seok Cheol Lee, Hyung Keun Kim, Ju Yeon Kang, Sree Samanvitha Kuppa, Jong Keun Seon

**Affiliations:** 1Department of Biomedical Sciences, Chonnam National University Medical School, Hwasun 58128, Jeollanam-do, Republic of Korea; jaanu.p2206@gmail.com (J.S.); sreesamanvitha95@gmail.com (S.S.K.); 2Department of Orthopaedic Surgery, Center for Joint Disease of Chonnam National University Hwasun Hospital, 322 Seoyang-ro, Hwasun 519-763, Jeollanam-do, Republic of Korea; iwannaseeu@naver.com (S.C.L.); chemokines@naver.com (H.K.K.); jy0194@naver.com (J.Y.K.); 3Korea Biomedical Materials and Devices Innovation Research Center, Chonnam National University Hospital, 42 Jebong-ro, Dong-gu, Gwangju 501-757, Republic of Korea

**Keywords:** cinnamaldehyde, articular inflammation, synovial inflammation, osteoarthritis

## Abstract

Inflammatory disorders encompass a range of conditions, including osteoarthritis (OA), characterized by the body’s heightened immune response to diverse stimuli. OA is a prevalent degenerative joint disease characterized by the progressive deterioration of joint cartilage and subchondral bone, leading to pain, limited mobility, and physical disability. Synovitis, the inflammation of the synovial membrane, is increasingly recognized as a critical factor in OA pathogenesis and progression. This study evaluates the therapeutic potential of cinnamaldehyde (CA), a bioactive compound derived from cinnamon, on synovial and articular inflammation in OA. Given CA’s established anti-inflammatory, antioxidant, and antibacterial properties, this research explores its specific impact on OA and synovitis. The cytotoxicity of CA was assessed using a CCK-8 assay in human IL-1β pretreated chondrocytes and synoviocytes, which serve as in vitro models of OA and synovitis. The study further examined the effects of CA on the expression of proinflammatory cytokines, including IL-6, COX-2, and TNF-α, utilizing multiple analytical techniques. Additionally, the production of matrix metalloproteinases (MMP-3 and MMP-13) and the activation of the NF-κB signaling pathway, particularly the phosphorylation of p65 (pp65), were investigated. The role of the NF-κB inhibitor 5HPP-33 and its downstream effects on gene expression, including COX-2 and IL-6, as well as the MAPK pathway components (p38, ERK, and JNK), were also explored. An MEK inhibitor (U0126) was employed to assess its downstream impact on COX-2 and IL-6 expressions. The results demonstrated that CA significantly inhibited the expression of proinflammatory cytokines and suppressed NF-κB activation in IL-1β pretreated chondrocytes and synoviocytes. These findings suggest that CA, in a dose-dependent manner, may serve as an effective therapeutic agent for preventing OA and synovitis, offering valuable insights into its potential role in managing synovial inflammation and OA.

## 1. Introduction

Inflammatory conditions represent a diverse group of disorders characterized by the body’s immune response to various stimuli, often resulting in tissue damage and chronic pain. Among these, osteoarthritis (OA) stands out as a prevalent and debilitating form of arthritis, characterized by the progressive degeneration of joint cartilage and subchondral bone, leading to pain, stiffness, and impaired mobility. It is the most common form of arthritis worldwide, primarily associated with aging, although it can also result from joint injury or obesity [1]. The disease significantly impacts the quality of life of affected individuals and imposes a substantial burden on healthcare systems. The pathophysiology of OA is complex, involving the breakdown of articular cartilage, alterations in subchondral bone, low-grade synovial inflammation, and changes in periarticular soft tissues. These pathological processes collectively contribute to joint pain and functional impairment [2]. Synovial inflammation, or synovitis, is a critical component in the pathogenesis and progression of OA [3]. In OA-affected joints, the synovium undergoes several pathological changes, including hyperplasia, increased vascularity, and infiltration by immune cells, particularly macrophages and T cells. These inflammatory alterations result in the production and release of proinflammatory mediators, such as cytokines (e.g., IL-1β, TNF-α), chemokines, and matrix-degrading enzymes, including matrix metalloproteinases (MMPs). These mediators diffuse into synovial fluid and cartilage, exacerbating inflammation, promoting cartilage degradation, and altering chondrocyte function [4]. The synovial lining in OA is characterized by hyperplasia, immune cell infiltration, neoangiogenesis, and fibrosis, all of which contribute to joint dysfunction and pain. The inflammatory process is driven by the activation of Toll-like receptors and the complement cascade in response to degradation products from cartilage and other joint tissues, leading to the release of cytokines and chemokines that exert catabolic effects on chondrocytes. This accelerates cartilage degradation and perpetuates OA progression [5]. The severity of synovitis has been correlated with both radiographic progression and pain in OA, underscoring its clinical significance. Moreover, synovial inflammation establishes a vicious cycle in which cartilage breakdown products further stimulate synovial inflammation, thereby perpetuating the disease process. Consequently, understanding and targeting synovial inflammation has emerged as a pivotal focus in the development of new therapeutic strategies for OA [6]. Current OA treatments are predominantly centered on symptom management, with limited options for disease modification. There is an urgent need for innovative therapies that can effectively target the underlying inflammatory processes and potentially slow or halt disease progression. In this context, natural compounds have garnered increasing attention for their anti-inflammatory and chondroprotective properties.

Cinnamon is a complex medicinal plant known for its diverse bioactive constituents, including essential oils and derivatives such as cinnamaldehyde (CA), cinnamic acid, and cinnamate. Among these, CA is the predominant compound, playing a critical role in cinnamon’s characteristic fragrance and its wide range of biological activities. Extensive research has demonstrated that CA exhibits significant anti-inflammatory, antioxidant, antimicrobial, and anticancer properties [7]. Notably, CA has garnered attention for its potential anti-inflammatory and antihyperglycemic effects, as well as its therapeutic implications in various disease contexts [8]. The essential oil derived from cinnamon bark is composed of approximately 90% CA, a naturally synthesized compound via the shikimate pathway [9]. Recent studies have highlighted CA’s potential in mitigating interleukin-1β (IL-1β)-induced chondrocyte and synovial inflammation, crucial factors in the pathogenesis of OA. Through modulation of key signaling pathways, including the nuclear factor-kappa B (NF-κB) and mitogen-activated protein kinase (MAPK) pathways, CA has been shown to suppress proinflammatory mediators and attenuate cartilage degradation [10]. These findings underscore the importance of investigating CA’s multifaceted role in OA pathophysiology. The current research aims to provide a comprehensive analysis of CA’s protective effects on chondrocytes and cartilage, particularly its capacity to inhibit inflammation and reduce cartilage degeneration. By elucidating the molecular mechanisms through which CA exerts its anti-inflammatory properties, this study seeks to offer new insights into potential therapeutic strategies for OA management. Given CA’s diverse pharmacological properties, including its anti-inflammatory, antioxidant, and potential anticancer effects, there is a compelling rationale for further exploration of its broader impact on joint health and disease. The findings of this research may contribute to the development of more effective, targeted therapies that address both synovial inflammation and cartilage preservation in OA, ultimately advancing the field of osteoarthritis treatment.

## 2. Results

### 2.1. Effect of Cinnamaldehyde on Chondrocyte and Synoviocyte Viability

The cytotoxicity of CA on chondrocytes and synoviocytes was assessed using the CCK-8 assay. Cells were exposed to increasing concentrations of CA (0.5, 5, and 50 μM) for 24 h. As shown in Figure 1A,B, the cellular cytotoxicity in response to CA treatment was evaluated. The results demonstrated that the viability of both chondrocytes and synoviocytes remained stable at CA concentrations up to 50 μM, with no significant reduction in cell viability observed at these concentrations. This indicates that CA is well-tolerated by these cell types within this range. However, a decline in cell viability was observed at 50 μM, with a more pronounced reduction at higher concentrations, a trend that was consistent for both cell types. Figure 1C,D illustrates the cellular viability of chondrocytes and synoviocytes when treated with IL-1β, simulating inflammatory conditions. This comparison between the effects of CA and an inflammatory environment on cell viability underscores the critical importance of dosage in therapeutic applications of CA, as higher concentrations may lead to reduced cell viability and potential cytotoxicity.

### 2.2. Morphological Analysis of Chondrocytes and Synoviocytes

Light microscopy of the chondrocyte cell line revealed a departure from the typical rounded morphology observed in native cartilage, with cells exhibiting an elongated structure. The nuclei were distinctly visible within these elongated cells, reflecting a morphological change indicative of the dynamic nature of chondrocytes. This alteration in cell shape and structure underscores the influence of the cellular environment on chondrocyte function. To further validate the chondrocyte phenotype and evaluate collagen content, we employed confocal microscopy. As shown in Figure 2A, the confocal images demonstrate the abundant collagen content, a hallmark of chondrocytes, thus providing additional confirmation of the chondrocytic nature of the cell line. Similarly, light microscopy of type B synoviocytes revealed a distinct, elongated, fibroblastic morphology, consistent with the typical appearance of fibroblast-like synoviocytes within the synovial membrane. To verify the synoviocyte phenotype, we utilized confocal microscopy to assess the expression of CD86, a surface marker associated with synoviocytes. The presence of CD86 on the cell surface, as illustrated in Figure 2B, further substantiates the identification of these cells as synoviocytes. These morphological assessments, in conjunction with specific marker analyses, affirm the identity and characteristics of both chondrocyte and synoviocyte cell lines. The elongated morphology of chondrocytes observed in culture, in contrast to their in vivo rounded shape, highlights the plasticity of these cells in response to environmental cues. Likewise, the fibroblastic morphology of synoviocytes is consistent with their established role within the synovial membrane.

### 2.3. CA Modulates MAPK Signaling Pathways in Chondrocytes and Synoviocytes

This study investigated the modulatory effects of CA on MAPK signaling pathways in chondrocytes and synoviocytes. Our findings revealed that CA exerted dose-dependent modulation on these pathways, with concentrations of 0.5, 5, and 50 μM demonstrating significant impacts. Specifically, in chondrocytes, a dose-dependent reduction in the levels of phosphorylated JNK, ERK, and p38 was observed at 0.5, 5, and 50 μM concentrations of CA, respectively. These reductions in phosphorylation levels suggest that the alterations in protein expression or localization, as observed through confocal microscopy, are consistent with the quantitative data obtained from Western blot analyses (Figure 3A–C). In synoviocytes, CA treatment similarly resulted in a significant decrease in phosphorylated p38 and pJNK levels, with a 15% reduction observed at the 0.5 μM concentration compared to untreated controls. This inhibitory effect was further amplified at higher concentrations, with 5 and 50 μM CA leading to more substantial reductions in pp38 and pJNK levels. Additionally, a dose-dependent decrease in phosphorylated ERK levels was noted in synoviocytes, where a significant reduction was detected at 0.5 μM, and further reductions were seen with 5 and 50 μM CA treatments. The consistency between confocal microscopy findings and Western blot data underscores the robustness of these observations (Figure 3D–F). Western blot analysis corroborated these findings, showing a clear dose-dependent decrease in the phosphorylation of p38, ERK1/2, and JNK in both chondrocytes and synoviocytes treated with CA. These results underscore CA’s efficacy in modulating MAPK signaling pathways, with the most pronounced effects manifesting at the highest concentration of 50 μM. The modulation of MAPK pathways by CA may underlie its anti-inflammatory and chondroprotective properties in the context of articular and synovial inflammation. To further elucidate the scope of CA’s anti-inflammatory action, we extended our investigation to other inflammation-related genes affected by CA treatment. Notably, the expressions of COX-2, cytokines, and MMPs were downregulated in a dose-dependent manner.

### 2.4. CA Dose-Dependently Reduces Inflammatory Mediators in Chondrocytes and Synoviocytes

This study evaluates the effects of CA on the expression of key inflammatory mediators in chondrocytes and synoviocytes, with a specific focus on the proinflammatory cytokines interleukin-6 (IL-6) and tumor necrosis factor-alpha (TNF-α), as well as cyclooxygenase-2 (COX-2) and matrix metalloproteinases (MMP-13 and MMP-3). These mediators are critical in the pathogenesis of inflammation and cartilage degradation in joint tissues. In chondrocytes, we observed a dose-dependent reduction in the expression of IL-6, TNF-α, COX-2, MMP-13, and MMP-3 at CA concentrations of 0.5 μM, 5 μM, and 50 μM. The reduction pattern was similar to that observed in synoviocytes, with the most significant decreases occurring at the highest CA concentration of 50 μM (Figure 4A,B). In synoviocytes, CA exhibited a similar dose-dependent inhibitory effect on all examined inflammatory mediators. At the lowest concentration of 0.5 μM, CA led to a 15–20% reduction in IL-6 expression, a significant reduction in TNF-α expression, a 10% reduction in COX-2 expression, and a notable decrease in MMP-13 and MMP-3 expression. These effects were progressively pronounced at 5 μM and 50 μM, with the most substantial reductions observed at 50 μM (Figure 4C,D).

These findings demonstrate that CA significantly attenuates the expression of proinflammatory cytokines (IL-6 and COX-2) in both chondrocytes and synoviocytes in a dose-dependent manner. The reductions were statistically significant across all tested concentrations, with the greatest effects observed at 50 μM CA. This comprehensive suppression of inflammatory mediators suggests that CA may have potential therapeutic applications in mitigating cartilage degradation and synovial inflammation in joint tissues.

### 2.5. CA Inhibits NF-κB Signaling and Reduces Inflammatory Markers in Chondrocytes and Synoviocytes via pp65 Suppression

Our study investigated the anti-inflammatory properties of CA, focusing on its effects on NF-κB signaling in chondrocytes and synoviocytes. We observed that treatment with CA at increasing concentrations (0.5, 5, and 50 μM) over 24 h resulted in a dose-dependent decrease in phosphorylated p65 (pp65) expression, a critical marker of NF-κB activation, across both cell types. Western blot analysis demonstrated the most substantial reduction at 50 μM CA, with pp65 levels showing a marked decrease in synoviocytes and a 70–80% reduction in chondrocytes compared to untreated controls. To further validate the specificity of CA’s action on the NF-κB pathway, we employed 5HPP-33, a recognized NF-κB inhibitor. Pre-treatment with 5HPP-33 (10 μM) effectively reversed the CA-induced reduction in pp65 expression in both cell types, indicating that CA’s effects are predominantly mediated through the NF-κB signaling pathway. Furthermore, we assessed the impact of CA on downstream inflammatory markers regulated by NF-κB. Western blot analysis revealed that treatment with CA (50 μM) significantly reduced the protein levels of the proinflammatory cytokines IL-6 and COX-2 in both chondrocytes and synoviocytes. This reduction was abrogated in cells pretreated with 5HPP-33, further supporting the NF-κB-dependent mechanism underlying CA’s anti-inflammatory activity. Collectively, these findings indicate that CA exerts its anti-inflammatory effects in chondrocytes (Figure 5A–D) and synoviocytes (Figure 5E–H) through dose-dependent inhibition of NF-κB activation, as evidenced by reduced pp65 expression and diminished levels of downstream inflammatory markers. The reversal of these effects by the NF-κB inhibitor 5HPP-33 confirms the pathway specificity of CA’s action, highlighting its potential as a targeted anti-inflammatory agent in conditions characterized by synovial and cartilage inflammation.

### 2.6. CA Inhibits ERK Signaling and Reduces Inflammatory Markers in Chondrocytes and Synoviocytes

We investigated the anti-inflammatory effects of CA on the ERK signaling pathway in chondrocytes and synoviocytes. Treatment with CA resulted in a significant inhibition of ERK signaling in both cell types. Western blot analysis revealed a marked reduction in phosphorylated ERK1/2 levels following CA treatment compared to untreated controls. This inhibition was comparable to that observed with the MEK1/2 inhibitor U0126, which is known to effectively block ERK phosphorylation. Quantitative Western blot analysis further demonstrated that CA treatment (0.5, 5, and 50 μM) led to a significant decrease in the expression of key inflammatory markers in both chondrocytes (Figure 6A,B) and synoviocytes (Figure 6C,D). Specifically, in osteoarthritic chondrocytes, CA treatment (50 μM) significantly reduced the expression of matrix metalloproteinases IL-6 and COX-2, both of which are involved in cartilage degradation and synovial inflammation. The reduction in IL-6 and COX-2 expression was similar to that observed with U0126 treatment (10 μM), suggesting that the anti-inflammatory effects of CA are mediated, at least in part, through the inhibition of ERK signaling. These findings suggest that CA may be a promising therapeutic agent for managing inflammation and cartilage degradation in OA, potentially acting through mechanisms similar to established ERK pathway inhibitors.

## 3. Discussion

Osteoarthritis is a prevalent and debilitating joint disorder that profoundly impacts the quality of life of millions of individuals globally. The global burden of OA is substantial, affecting over 250 million people, with incidence rates increasing annually [11]. According to the 2017 Global Burden of Disease (GBD) Study, OA accounts for approximately 1.0% of all disability-adjusted life years (DALYs), highlighting its significant contribution to global disability. The prevalence of OA varies by age, gender, and geographical location, with older women being particularly susceptible. In the United States, the age-standardized prevalence of OA has been reported at 6128.1 cases per 100,000 population, with similarly high prevalence rates observed in other regions, including Europe and Asia [12]. Notably, in regions of cartilage–synovium interfaces, particularly under inflammatory conditions, an increased presence of synoviocytes expressing both fibroblast and macrophage markers has been observed, suggesting a potential phenotypic shift or activation state [13]. Given the chronic nature of OA and the limitations of current pharmacological treatments, there is growing interest in the use of natural products (NPs) as alternative or complementary therapies. NPs derived from plants, animals, and microorganisms have demonstrated promising potential in preclinical and clinical studies for the treatment of OA. These products are believed to exert their therapeutic effects through various mechanisms, including the modulation of key signaling pathways such as NF-κB, MAPKs, PI3K/AKT, and sirtuin-1(SIRT1), which are involved in inflammation, anabolism, catabolism, and cell death [14]. Numerous clinical trials have substantiated the efficacy of various NPs in mitigating OA symptoms. For example, Sphaeralcea angustifolia has demonstrated the capacity to reduce pain, inflammation, and stiffness in OA patients without eliciting adverse reactions [15]. Similarly, Cucumis sativus extract has been effective in lowering Western Ontario and McMaster Universities Arthritis Index (WOMAC) and Visual Analogue Scale (VAS) scores in randomized controlled trials, indicating its potential to enhance joint function and alleviate pain. Additional NPs, such as hydroxytyrosol derived from olive leaves [16] and GCWB106 from Chrysanthemum zawadskii, have also exhibited significant therapeutic effects in clinical studies, further reinforcing the potential role of NPs in OA treatment [17]. The therapeutic effects of NPs in OA are primarily attributed to their ability to modulate molecular pathways implicated in the disease’s pathogenesis. A recent experimental study has confirmed that Duhuo Jisheng Decoction (DHJSD) plays a critical role in OA treatment by targeting the TLR4/MyD88/NF-κB signaling pathway, which is pivotal in the inflammatory processes associated with OA. The study employed a network pharmacology approach integrated with experimental validation, revealing that inhibition of this pathway significantly reduces OA-related inflammation. Notably, DHJSD administration resulted in alterations in miR-146a-5p and miR-34a-5p in cellular models, which synergistically contributed to OA therapy by enriching Toll-like receptor and NF-κB signaling pathways through miRNA-regulated gene pathways [18]. Furthermore, NPs such as curcumin and green tea extract have been shown to inhibit the NF-κB pathway, thereby reducing inflammation and preventing cartilage degradation [19]. Exosome therapy is also emerging as a promising strategy in OA management [20]. Recent studies indicate that exosomes derived from human bone marrow mesenchymal stem cells, when treated with cinnamaldehyde, exhibit anti-inflammatory effects in chondrocyte cells. Cinnamaldehyde, recognized for its anti-inflammatory properties, enhances the therapeutic potential of these exosomes by modulating key inflammatory pathways, including the NF-κB and p38-JNK pathways, which are critical in the inflammatory response associated with OA [21]. This combination has demonstrated potential in reducing inflammation and preserving chondrocytes, highlighting a novel approach to the treatment of inflammatory conditions such as OA [22]. Recent years have seen a growing interest in natural products as potential therapeutic agents for various diseases, including synovial and articular inflammation. CA, a bioactive compound derived from cinnamon, has demonstrated promising anti-inflammatory effects in joint tissues [23]. However, the precise mechanisms by which CA exerts its anti-inflammatory and potential tissue repair effects in joint tissues remain to be fully elucidated. Cinnamon, known as “Rou Gui” in Chinese, has been used for centuries in traditional Chinese medicine (TCM) to treat a variety of conditions, including digestive issues, colds, and respiratory problems [24]. It is also used to warm the body and improve circulation. Additionally, in Ayurvedic medicine, cinnamon is known as “Dalchini” and is used to treat several health conditions, including digestive problems, diabetes, and respiratory issues. It is valued for its warming properties and is often used in combination with other herbs [25]; in Unani medicine, a traditional system of healing that originated in Greece and was later adopted by Arab and Indian physicians, cinnamon is used to treat various ailments such as indigestion, flatulence, and colds. It is also believed to have anti-inflammatory properties [26]. In many folk medicine traditions, cinnamon is used for its anti-inflammatory, antioxidant, and antimicrobial properties. For example, it is used to treat sore throats, coughs, and other respiratory infections [23]. Our research suggests that CA significantly enhances chondrocyte viability in response to IL-1β treatment, indicating a protective effect on cartilage. Additionally, CA appears to modulate synoviocyte function, which may contribute to a reduction in synovial inflammation. Emerging evidence indicates that CA’s anti-inflammatory effects involve multiple pathways, including the inhibition of the TLR4/MyD88 signaling pathway in OA synovial fibroblasts, as well as the suppression of NF-κB and p38-JNK pathways in chondrocytes [27]. Furthermore, CA treatment has been shown to decrease the expression of proinflammatory cytokines such as IL-1β, IL-6, TNF-α, IL-23, and IL-17 in various arthritis models [28,29]. CA also suppresses HIF-1α activity by inhibiting succinate accumulation in inflammatory cells and reducing the expression of the succinate receptor GPR91. Additionally, CA inhibits NLRP3-derived IL-1β production, a critical driver of inflammation in rheumatoid arthritis [30]. Recent studies have further demonstrated that CA suppresses inflammation in chondrocytes by inducing the expression of microRNA-1285-5p and microRNA-140-5p, thereby ameliorating apoptosis and the inflammatory response in these cells [31]. The investigation of cinnamaldehyde-based self-nanoemulsion delivery systems (CA-SNEDDS) has revealed significant anti-inflammatory effects in a rat skin burn model, where CA-SNEDDS was shown to accelerate wound healing and exert antimicrobial, antioxidant, and anti-inflammatory effects. Specifically, CA-SNEDDS treatment led to a marked reduction in inflammatory markers and an enhancement of antioxidant biomarkers, such as superoxide dismutase (SOD) and catalase (CAT), compared to the untreated group. These findings suggest that CA-SNEDDS effectively reduces inflammation and promotes healing in burn injuries [32]. Moreover, CA treatment has been shown to downregulate the expression of COX-2 and other inflammatory markers in joint tissues. By targeting key inflammatory and catabolic pathways, natural products like CA may alleviate OA symptoms and potentially slow disease progression (Figure 7).

These investigations contribute to a more comprehensive understanding of CA’s role in managing synovial and articular inflammation, potentially paving the way for the development of novel, targeted therapies for OA and related joint disorders. This study underscores the potential of CA in mitigating synovial and articular inflammation in vitro. However, several challenges must be addressed to facilitate clinical translation. Ensuring the consistent quality and concentration of CA is essential for reproducible research, given the inherent variability in natural product composition that can lead to inconsistent outcomes. Further investigations are warranted to elucidate CA’s anti-inflammatory mechanisms, identify specific molecular targets, enhance bioavailability, and develop innovative delivery systems. Moreover, long-term clinical trials are necessary to evaluate the safety and efficacy of CA, and the identification of biomarkers for personalized treatment approaches remains a critical priority. Bridging the gap between preclinical findings and clinical efficacy presents a significant challenge. Additionally, fulfilling regulatory requirements and resolving intellectual property issues are imperative for successful commercialization. Future research efforts hold the promise of advancing CA’s therapeutic potential, potentially leading to novel treatment strategies for OA. In conclusion, this study demonstrates the anti-inflammatory and potentially chondroprotective effects of CA on synovial and articular inflammation, primarily through the modulation of key inflammatory pathways, particularly the MAPK and NF-κB signaling cascades. Our investigation into the ERK pathway utilized the specific MAPK inhibitor U0126 to elucidate the mechanisms through which CA reduces inflammatory mediators. Western blot analysis substantiated the functional involvement of pERK, revealing apparent inhibition by CA, as evidenced by reduced pERK intensity in chondrocytes and synoviocytes cells. Notably, the addition of the ERK-specific inhibitor U0126 resulted in significant reductions in the expression levels of COX-2 and IL-6, highlighting its potential role in mediating CA’s anti-inflammatory effects on chondrocytes and synoviocytes. While our initial focus was on the ERK pathway as a representative of MAPK signaling, we acknowledge the need for further studies employing additional MAPK pathway inhibitors to provide a more comprehensive understanding of CA’s effects on inflammatory mediators in osteoarthritis. Treatment with CA results in the downregulation of proinflammatory cytokines such as IL-6 and TNF-α, as well as the suppression of COX-2 and inflammatory mediators like MMP-3 and MMP-13. Enzyme-linked immunosorbent assay (ELISA) was not conducted in this study due to the limited sensitivity of the technique for detecting the low concentrations of cytokines typically present in our experimental conditions. Instead, we opted for more sensitive methods such as Polymerase chain reaction (PCR) and Western blotting to assess cytokine expression and protein levels, respectively. These findings contribute to the expanding body of research supporting the therapeutic potential of natural products in the management of inflammatory joint diseases.

## 4. Materials and Methods

### 4.1. Antibodies and Reagents

Cell culture media, including 1 × Dulbecco’s Modified Eagle’s Medium (DMEM), penicillin-streptomycin (pen-strep), and fetal bovine serum (FBS), were acquired from Gibco (Thermo Fisher Scientific, Waltham, MA, USA). Human recombinant IL-1β was sourced from R&D Systems (Minneapolis, MN, USA) and dissolved in phosphate buffered saline (PBS) containing 0.5% bovine serum albumin (BSA). Antibodies specific to COX-2 (Abcam, Boston, MA, USA), TNF-α (Abcam, Boston, MA, USA), IL-6 (cell signaling, Danvers, MA, USA), MMP-3 (BioLegend, San Diego, CA, USA), MMP-13 (Bioss, Woburn, MA, USA), p65 (cell signaling, Danvers, MA, USA), pp65 (cell signaling, Danvers, MA, USA), ERK (cell signaling, Danvers, MA, USA), pERK (cell signaling, Danvers, MA, USA), p38 (cell signaling, Danvers, MA, USA), pp38 (cell signaling, Danvers, MA, USA), JNK (Bioss, Woburn, MA, USA), and pJNK (cell signaling, Danvers, MA, USA) were procured. Additionally, goat anti-mouse IgG (ZyMax, Thermo Fisher Scientific, Waltham, MA, USA) and a conjugated goat anti-rabbit secondary antibody (H + L) (Novex Life Technologies, Thermo Fisher Scientific, Waltham, MA, USA) were purchased. CA (≥95% purity, W228613) was obtained from Sigma-Aldrich (St. Louis, MO, USA) and dissolved in dimethyl sulfoxide (DMSO) to prepare a stock solution.

### 4.2. Source of Cells and Cell Culture

The human articular chondrocyte cell line and human synoviocyte cell line were obtained from ScienCell (#4650, #4700). The cells were cultured in 1 × DMEM, supplemented with 1% pen-strep and 10% heat-inactivated FBS, at 37 °C in a 5% CO_2_ atmosphere. Cells were transferred to a culture plate and sub-cultured when the cell density reached 80%. Chondrocytes from passages 3 to 6 were utilized for experiments. It should be noted that no donor criteria are provided for chondrocyte cells; further details on the characterization of HC-a cells and synoviocytes can be obtained by contacting the supplier, ScienCell [33,34]. For certain experiments, passages 3–4 were used, with cells maintained in 1 × DMEM supplemented with 1% pen-strep and 10% heat-inactivated exosome-free serum at 37 °C in a 5% CO_2_ atmosphere.

### 4.3. Evaluation of Cell Viability

Cell viability was assessed using the CCK-8 assay kit (Dojindo, Japan) following treatment with IL-1β and CA. Cells were seeded in 96-well plates at a density of 1 × 10^4^ cells/well, stimulated with IL-1β for 24 h, and subsequently treated with or without CA at concentrations of 0.5, 5, and 50 µM for 24 h in both chondrocytes and synoviocytes. After the appropriate treatment according to experimental grouping, 10 µL of CCK-8 solution was added to each well and incubated for 2 h at room temperature. Absorbance at 450 nm was measured using a BioTek Synergy HTX multimode plate reader (Agilent Technologies, Santa Clara, CA, USA) to quantify the dye formed. The CCK-8 assay is a colorimetric method used for evaluating cell proliferation and cytotoxicity, offering a ready-to-use solution that requires no pre-mixing of components. This assay provides a simple, rapid, reliable, and sensitive measurement of cell viability and cytotoxicity across various chemical treatments. Notably, the CCK-8 assay exhibits higher detection sensitivity compared to other tetrazolium salts such as MTT, XTT, MTS, or WST-1.

### 4.4. Role of CA in IL-1β-Induced Chondrocyte Inflammation and Synovial Inflammation

Human articular chondrocytes were seeded at 5 × 10^5^ cells per well into 65 mm plates containing 1X DMEM supplemented with 10% FBS and 1% penicillin/streptomycin and cultured at 37 °C under 5% CO_2_ for 24 h. Upon reaching confluence, the chondrocytes were induced with 10 ng/mL IL-1β to initiate inflammation. CA was reconstituted in PBS at concentrations of 0.5, 5, and 50 µM. Post-treatment with CA was conducted after exposure to the proinflammatory cytokine IL-1β, rather than pre-treatment, to better mimic clinical conditions. Following 24 h of incubation, total RNA and total protein were extracted from chondrocytes and synoviocytes using TRIzol reagent and Radioimmunoprecipitation assay (RIPA) buffer, respectively.

### 4.5. PCR

Total RNA was extracted, and 0.5 μg of total RNA was used for first-strand complementary DNA (cDNA) synthesis. Aliquots of the RNA were reverse-transcribed in 20 μL of buffer containing 200 U of M-MLV reverse transcriptase, 0.25 mM DTT, and 250 μM of each dATP, dCTP, dGTP, and dTTP. The reverse transcription was performed under the following conditions: an initial incubation at room temperature for 10 min, followed by 12 cycles of 25 °C for 30 s, 45 °C for 4 min, and 55 °C for 30 s, concluding with a final step at 95 °C for 5 min and a 4 °C hold, using a GeneAmp PCR System 2700 (Applied Biosystems, Foster City, CA, USA). Subsequently, aliquots of the cDNA were amplified using the AccuPower^®^ GreenStar PCR premix (Bioneer Co., Daejeon, Republic of Korea) in the MyGenie™ 96/384 thermal cycler system (Bioneer Co., Daejeon, Republic of Korea). Gene expression assays were performed for IL-6, TNF-α, COX 2, MMP-3, MMP-13, and GAPDH, with expression levels normalized to and assessed against the expression of GAPDH (Table 1).

### 4.6. Whole-Cell Lysate Preparation and Western Blot Analysis

Following treatment, cells were lysed using radioimmunoprecipitation assay (RIPA) lysis buffer supplemented with protease and phosphatase inhibitor cocktails. Protein concentrations from the whole-cell lysates, subjected to CA treatment in a dose-dependent manner, were quantified using the bicinchoninic acid (BCA) protein assay kit (Thermo Scientific, Waltham, MA, USA). Standards and samples were loaded into a 96-well plate, mixed with the working reagent, and incubated at 37 °C for 30 min. Absorbance was measured at 562 nm using a spectrophotometer. Protein samples (5 µg) were then incubated at 95 °C for 5 min, separated via polyacrylamide gel electrophoresis (SDS-PAGE), and transferred onto polyvinylidene difluoride (PVDF) membranes. The membranes were blocked with 5% skim milk (DifcoTM, Becton Drive, Franklin Lakes, NJ, USA) for 90 min at 4 °C, followed by overnight incubation with primary antibodies specific for IL-6, TNF-α, COX-2, MMP-3, MMP-13, p65, pp65, ERK, pERK, p38, pp38, JNK, pJNK, and the housekeeping gene α-Tubulin. After three 10 min washes with TBS-Tween20 (TBST), the membranes were incubated with Horse-radish peroxidase (HRP)-conjugated goat anti-mouse IgG (1:5000) and goat anti-rabbit IgG (H + L) highly cross-adsorbed secondary antibodies for 2 h at 4 °C. The membranes were subsequently washed three times with TBST, each wash lasting 10 min, to remove unbound or nonspecifically bound proteins. The membranes were then incubated with chemiluminescent substrates, with incubation times varying between 1 and 5 min depending on the substrate sensitivity and the target protein. The chemiluminescent reaction produced light detectable by a luminometer, enabling quantification of the protein present on the membrane. Blots were visualized using an enhanced chemiluminescence (ECL) system (Millipore, Bedford, MA, USA), and band intensities were quantified using ImageJ Version 1.54j software (National Institutes of Health, Bethesda, MD, USA).

### 4.7. Inhibitor Treatment

Chondrocytes and synoviocytes (1 × 10^6^/mL) were cultured for 3 days in 10% DMEM and FBS, supplemented with penicillin (100 U/mL), prior to their use in the experiments. Following this, the cells were serum-starved for 6 h to equilibrate all cell populations. Pre-treatment was carried out with either 10 μM of the NF-κB signaling inhibitor 5HPP-33 (Calbiochem, La Jolla, CA, USA) or 10 μM of the MEK 1/2 kinase inhibitor U0126 (Cell Signaling, Danvers, MA, USA) for 2 h. Subsequently, the cells were exposed to IL-1β (10 ng/mL) in the presence or absence of CA (0.5, 5, and 50 μM) for 24 h.

### 4.8. Confocal Microscopy

Chondrocyte cells cultured on coverslips were fixed with 4% paraformaldehyde in PBS for 15 min and permeabilized with 0.1% Triton X-100 for 15 min. The cells were then blocked with 1% BSA in 0.1% Triton X-100 for 60 min. Coverslips were incubated overnight at 4 °C with primary antibodies against pp38, pERK, pJNK, and pp65 at a 1:200 dilution, followed by secondary antibodies at a 1:400 dilution. After washing with PBS, the cells were stained with phalloidin for 15 min, washed again with PBS, and mounted using DAPI and a mounting medium. Fluorescence micrographs were acquired using a Carl Zeiss microscopy GmbH (ZEISS LSM 980) at 40× magnification to capture the desired images and analyze colocalization patterns. Data analysis was performed using ZEN 3.2 (Blue edition) software.

### 4.9. Data Collection and Statistical Analysis

Statistical analyses were conducted using GraphPad Prism software, version 9.0 (GraphPad Software, San Diego, CA, USA). For multiple group comparisons involving parametric data, one- or two-way ANOVA was performed using GraphPad Prism software, version 9.0 (GraphPad Software, San Diego, CA, USA). A *p*-value of < 0.05 was considered statistically significant. Data are presented as the mean ± standard deviation (*n* = 3). Statistical significance is indicated as follows: ns = non-significant, * *p* < 0.05, ** *p* < 0.01, *** *p* < 0.001, and **** *p* < 0.0001, compared with the control group.

## 5. Conclusions

In conclusion, this study demonstrates the anti-inflammatory and potentially chondroprotective effects of CA on synovial and articular inflammation, primarily through the modulation of key inflammatory pathways, particularly the MAPK and NF-κB signaling cascades. Treatment with CA results in the downregulation of proinflammatory cytokines such as IL-6 and TNF-α, as well as the suppression of COX-2 and inflammatory mediators like MMP-3 and MMP-13. These findings contribute to the expanding body of research supporting the therapeutic potential of natural products in the management of inflammatory joint diseases.

## Figures and Tables

**Figure 1 ijms-25-12914-f001:**
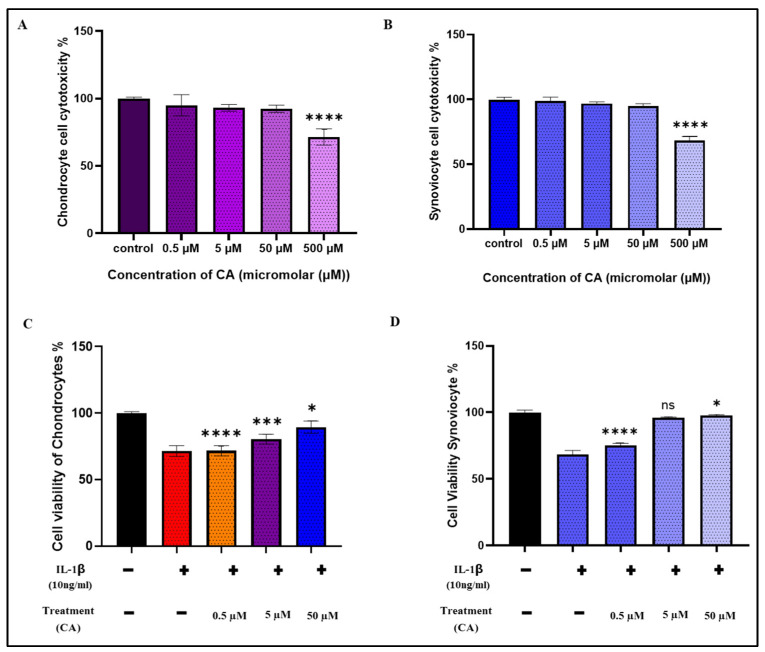
(**A**) Cell cytotoxicity of chondrocytes and (**B**) cell cytotoxicity of synoviocytes with the treatment of cinnamaldehyde in a dose-dependent manner (0.5, 5, 50, and 500 μM) for 24 h. (**C**) Cell viability of chondrocytes and (**D**) cell viability of synoviocytes with the treatment IL-1β prior to cinnamaldehyde in a dose-dependent manner (0.5, 5, 50, and 500 μM) for 24 h. Data are presented as the mean ± standard deviation (*n* = 3). ns = non-significant, * *p* < 0.01, *** *p* < 0.001, and **** *p* < 0.0001 compared with the control group.

**Figure 2 ijms-25-12914-f002:**
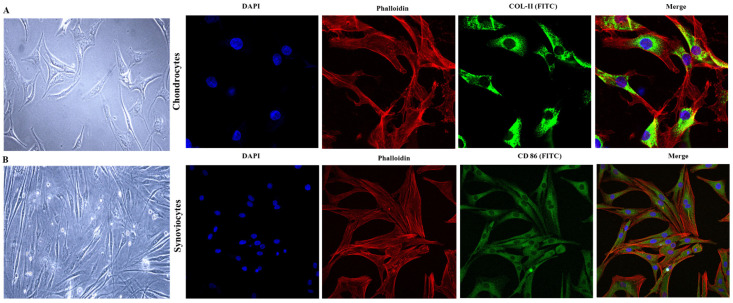
(**A**) Cell microscopy of chondrocytes followed by confocal microscopy of COL-II surface marker expression, and (**B**) cell microscopy of synoviocytes followed by confocal microscopy of CD 86 surface marker expression at 40X magnification (scale bar = 100 μM). (DAPI (blue)-Nucleus, Phalloidin (red)-F-actin, and COL-II and CD-86 (green)).

**Figure 3 ijms-25-12914-f003:**
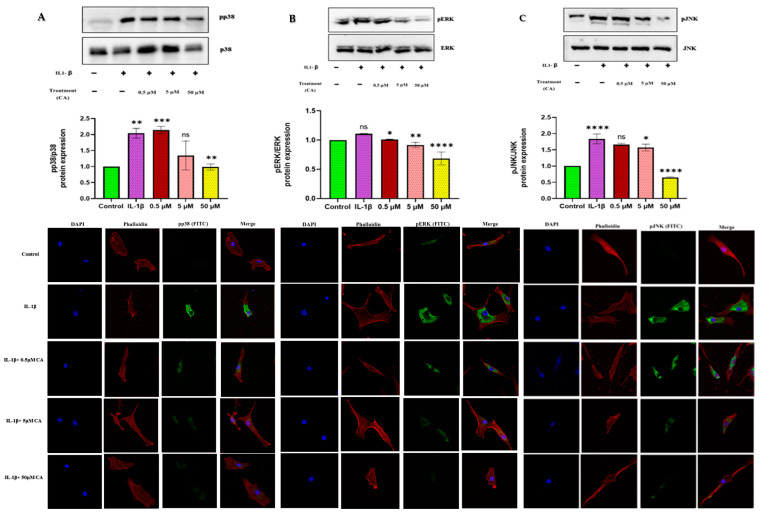
(**A**–**C**) Protein expression of MAPK pathways (p38, pERK, and pJNK) in chondrocytes followed by qualitative analysis and confocal microscopy at 40× magnification in a dose-dependent manner (DAPI (blue)-Nucleus, Phalloidin (red)-F-actin, and p38, pERK, and pJNK (green)). (**D**–**F**) Protein expression of MAPK pathways (p38, pERK, and pJNK) in synoviocytes followed by qualitative analysis and confocal microscopy at 40× magnification in a dose-dependent manner (DAPI (blue)-Nucleus, Phalloidin (red)-F-actin, and p38, pERK, and pJNK (green)) (scale bar = 100 μM). Data are presented as the mean ± standard deviation (*n* = 3). ns = non-significant, * *p* < 0.05, ** *p* < 0.01, *** *p* < 0.001, and **** *p* < 0.0001 compared with the control group.

**Figure 4 ijms-25-12914-f004:**
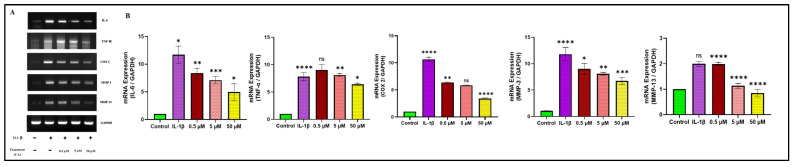
(**A**–**D**) Gene expression and protein expression of inflammatory mediators in chondrocytes, followed by qualitative analysis. (**E**–**H**) Gene expression and protein expression of inflammatory mediators in synoviocytes, followed by qualitative analysis. Data are presented as the mean ± standard deviation (*n* = 3). ns = non-significant, * *p* < 0.05, ** *p* < 0.01, *** *p* < 0.001, and **** *p* < 0.0001 compared with the control group.

**Figure 5 ijms-25-12914-f005:**
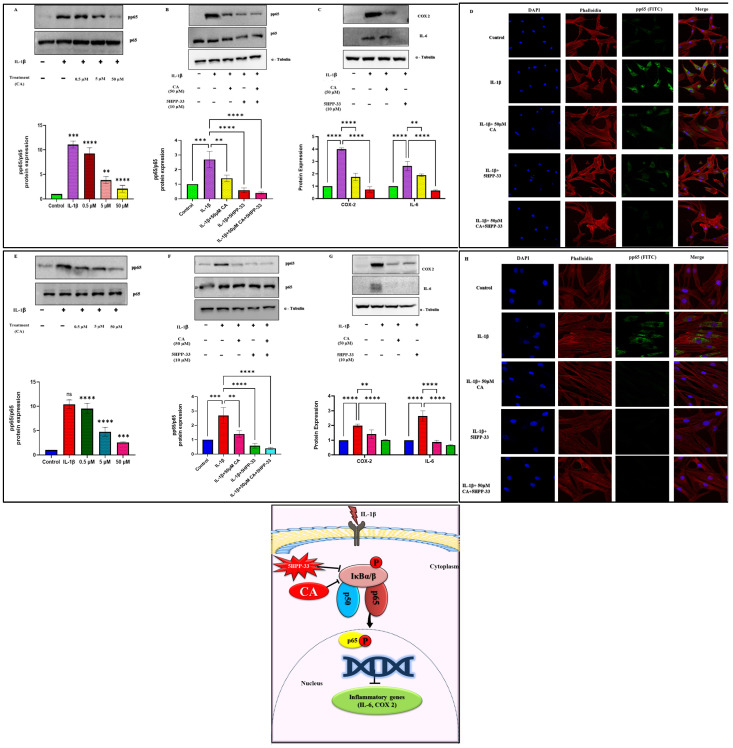
(**A**) Protein expression of NF-κB pathways (pp65) in chondrocytes in a dose-dependent manner, followed by qualitative analysis. (**B**) Protein expression of NF-κB pathways (pp65) in the presence and absence of NF-κB inhibitor (5HPP-33) in chondrocytes. (**C**) Comparison of the downstream mediators of inflammation (COX-2, IL-6) in the presence of an inhibitor and 50 μM of CA in chondrocytes, followed by qualitative analysis. (**D**) Confocal microscopy (pp65) at 40× magnification in the presence and absence of NF-κB inhibitor (5HPP-33) in chondrocytes (Figure) (DAPI (blue)-Nucleus, Phalloidin (red)-F-actin, and pp65 (green)). (**E**) Protein expression of NF-κB pathways (pp65) in synoviocytes in a dose-dependent manner, followed by qualitative analysis. (**F**) Protein expression of NF-κB pathways (pp65) in the presence and absence of NF-κB inhibitor (5HPP-33) in synoviocytes. (**G**) Comparison of the downstream mediators of inflammation (COX-2, IL-6) in the presence of NF-κB inhibitor and 50 μM of CA in synoviocytes, followed by qualitative analysis. (**H**) Confocal microscopy (pp65) at 40× magnification in the presence and absence of NF-κB inhibitor (5HPP-33) in synoviocytes. (DAPI (blue)-Nucleus, Phalloidin (red)-F-actin, and pp65 (green)). (Scale bar = 100 μM). Data are presented as the mean ± standard deviation (*n* = 3). ns = non-significant, ** *p* < 0.01, *** *p* < 0.001, and **** *p* < 0.0001 compared with the control group.

**Figure 6 ijms-25-12914-f006:**
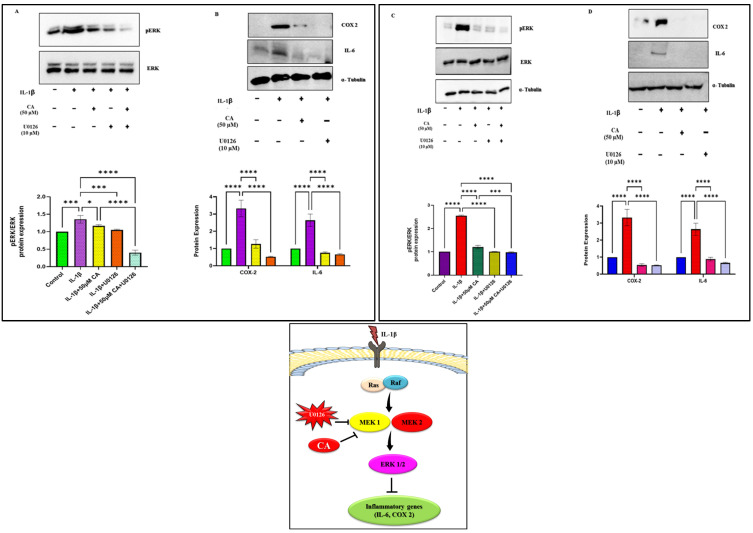
(**A**) Protein expression of MAPK pathways (pERK phosphorylation) in the presence of MEK1/2 inhibitor (U0126) in chondrocytes, followed by qualitative analysis. (**B**) Comparison of the downstream mediators of inflammation (COX-2, IL-6) in the presence of MEK 1/2 inhibitor and 50 μM of CA in synoviocytes, followed by qualitative analysis. (**C**) Protein expression pERK phosphorylation in the presence of MEK1/2 inhibitor (U0126) in synoviocytes, followed by qualitative analysis. (**D**) Comparison of the downstream mediators of inflammation (COX-2, IL-6) in the presence of MEK 1/2 inhibitor and 50 μM of CA in synoviocytes, followed by qualitative analysis. The figure represents the activity of CA and Inhibitor role in the ERK pathway inhibition. Data are presented as the mean ± standard deviation (*n* = 3). ns = non-significant, * *p* < 0.05, *** *p* < 0.001, and **** *p* < 0.0001 compared with the control group.

**Figure 7 ijms-25-12914-f007:**
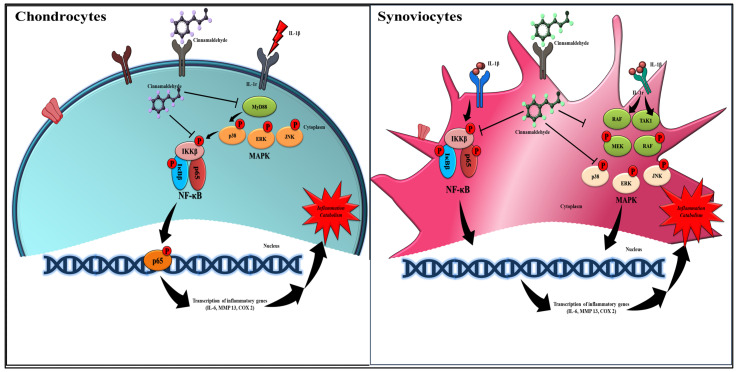
A schematic explanation of the mechanism by which cinnamaldehyde (CA) upregulates *p*-NF-κB (*p*-p65) expression in chondrocytes and synoviocytes because of IL-1β stimulation. IL-1β alters the transcriptional activity of MAPK by decreasing the expression of downstream mediators such as IL-6, MMP13, and COX 2. The effect of CA is exerted through a reduction in pNF-κB (*p*-p65) expression and the pMAPK (pp38, pERK, pJNK) expression in both cell types. (→ denotes the process of activation and ⊥ denotes the inhibition of the activity).

**Table 1 ijms-25-12914-t001:** Human Primer Sequences Used in This Study.

Primer Name	Forward Primer	Reverse Primer
Human IL-6	5′-GGATGCTTCCAATCTGGATTCAATGAG-3′	5′-CGCAGAATGAGATGAGTTGTCATGTCC-3′
Human TNF-α	5′-AGGCGGTGCTTGTTCCTC-3′	5′-GTTCGAGAAGATGATCTGACTGCC-3′
Human MMP3	5′-GGCAGTTTGCTCAGCCTATC-3′	5′-GTCACCTCCAATCCAAGGAA-3′
Human MMP13	5′-GATGAAGACCCCAACCCTAAA-3′	5′-CTGGCCAAAATGATTTCGTTA-3′
Human COX-2	5′-TTC AAATGAGATTGTGGGAAA-3′	5′-AGATCATCTCTGCCTGAGTATCTT-3′
Human GAPDH	5′-ACCACAGTCCATGCCATCAC-3′	5′-TCCACCACCCTGTTGCTGTA-3′

## Data Availability

All data generated or analyzed in this study are included in the article.

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
