# Peer review of "Cinnamaldehyde-Mediated Suppression of MMP-13, COX-2, and IL-6 Through MAPK and NF-κB Signaling Inhibition in Chondrocytes and Synoviocytes Under Inflammatory Conditions"

_ijms, 2024, doi:10.3390/ijms252312914_

Round 1

Reviewer 1 Report

Comments and Suggestions for Authors

1. Why did you not try cell viability assay between 50 and 500 uM?

2.  Paragraphs should be justified.

3. In figure 2 there is no error bar.  Contrast in light microscopy  figure should be increase. Which is the input of this figure? Why it is important?

4. In figure 3, figure legend is up of the figure, please correct. Too many pixels on the figures. Confocal images are too little. There is no error bars. What are the different colors. This figure is awful.

5. Same with figure 4. Too much information, too little the  figures.

6. Same for figure 5.

7. A final diagram should be include.

8. Is there any tradicional medicine using cinnamon? please include.

Author Response

Dear Reviewer,

We appreciate your insightful comments and valuable suggestions regarding our manuscript titled Cinnamaldehyde-mediated suppression of MMP-13, COX-2, and IL-6 through MAPK and NF-κB signaling inhibition in chondrocytes and synoviocytes on inflammatory condition. We sincerely appreciate your time, effort, and expertise in evaluating our manuscript. Your invaluable contributions have undoubtedly enhanced the quality of our work. We are grateful for the opportunity to benefit from your expertise and insights.

Thank you once again. We look forward to any further suggestions or guidance you may have as we continue to advance our research. In response to your valuable suggestions, we have made the following revisions:

Point 1. Why did you not try cell viability assay between 50 and 500 uM?

Response 1: Thank you for your detailed feedback on our manuscript. Our primary focus was on identifying the therapeutic window where CA is effective without causing significant cytotoxicity. The data from the concentrations tested (0.5, 5, and 50 μM) provided sufficient evidence to support the anti-inflammatory and chondroprotective effects of CA within a safe and effective dose range.

Point 2.  Paragraphs should be justified.

Response 2: Thank you for pointing out the issues. As the paragraph justification, Yes, it was done.

Point 3. In figure 2 there is no error bar.  Contrast in light microscopy  figure should be increase. Which is the input of this figure? Why it is important?

Response 3: We thank you for highlighting the need for error bars and will include them in the revised figures to indicate variability and statistical significance

Figure 2 appears to be a morphological analysis using light microscopy and confocal microscopy. Since these figures are typically used to illustrate the morphology and marker expression of cells rather than quantitative data, error bars are not usually included. We adjusted the contrast of the light microscopy images to improve the visibility of the cellular structures, ensuring that the morphology of chondrocytes and synoviocytes is clearly evident

Point 4. In figure 3, figure legend is up of the figure, please correct. Too many pixels on the figures. Confocal images are too little. There is no error bars. What are the different colors. This figure is awful.

Response 4: Thank you for suggesting the inclusion of color keys; we will make sure to add these to the figures for better interpretation.

We optimized the image resolution to reduce pixelation and improve the overall quality of the figures. This will involve using higher-resolution images or adjusting the figure settings to minimize pixelation. We ensured that the figure legend is placed below the figure, following standard formatting conventions. We included a color key or legend within the figure to explain the different colors used in the confocal images. (DAPI(Blue)-Nucleus, Phalloidin (Red)-F-actin, pp65 (Green))

Point 5. Same with figure 4. Too much information, too little the  figures.

Response 5: Thank you for your insightful comments; we carefully addressed each point to enhance the overall quality of our figures and manuscript. We revised Figure 4 to address all the mentioned issues, ensuring that it is clear, well-organized, and of high quality.

Point 6. Same for figure 5.

Response 6: We appreciate your detailed feedback and will revise the figures to address all the mentioned issues, ensuring they are clear, well-organized, and of high quality. We revised Figure 4 to address all the mentioned issues, ensuring that it is clear, well-organized, and of high quality.

Point 7. A final diagram should be include.

Response 7: We are grateful for your thorough review and valuable feedback, which will help us improve the quality and clarity of our manuscript. Yes, it is included in the file.

Point 8. Is there any tradicional medicine using cinnamon? please include.

Response 8: Thankyou so much for your valuable insights as you have suggested we have included the traditional as well as other medicinal methods in cinnamon.

Cinnamon, known as "Rou Gui" in Chinese, has been used for centuries in Traditional Chinese Medicine (TCM) to treat a variety of conditions, including digestive issues, colds, and respiratory problems. It is also used to warm the body and improve circulation.

In Ayurvedic medicine, cinnamon is known as "Dalchini" and is used to treat several health conditions, including digestive problems, diabetes, and respiratory issues. It is valued for its warming properties and is often used in combination with other herbs.

In Unani medicine, a traditional system of healing that originated in Greece and was later adopted by Arab and Indian physicians, cinnamon is used to treat various ailments such as indigestion, flatulence, and colds. It is also believed to have anti-inflammatory properties.

In many folk medicine traditions, cinnamon is used for its anti-inflammatory, antioxidant, and antimicrobial properties. For example, it is used to treat sore throats, coughs, and other respiratory infections.

Reviewer 2 Report

Comments and Suggestions for Authors

This study has some major issues:

1.     In Figures 1C and 1D, how much IL-1β was used to treat the cells? Were the cells treated with IL-1β + CA at concentrations ranging from 5 to 50 μM? If so, please re-label the figure, as the current labeling is very confusing.

2.     Please revise the font and labeling in all figures. The current font and figure sizes are too small, making them difficult for readers to interpret.

3.     For assessing pro-inflammatory markers, why did the authors not include any ELISA data?

4.     Why does the illustration in Figure 5 focus on the NF-κB inhibitor, and the illustration in Figure 6 focus on U0126? The authors did not include any illustration describing how CA inhibits the pro-inflammatory signaling pathway in chondrocytes and synoviocytes in this manuscript.

5.     In Figure 6, why did the authors use only the ERK inhibitor U0126? Based on the data in Figure 3, CA also effectively reduced IL-1β-induced p-JNK and p-p38 levels.

6.     Why did the authors use 10 μM of 5HPP-33 and U0126? Is there any data showing that this is the optimal dose?

Comments on the Quality of English Language

Acceptable

Author Response

Dear Reviewer,

We appreciate your insightful comments and valuable suggestions regarding our manuscript titled Cinnamaldehyde-mediated suppression of MMP-13, COX-2, and IL-6 through MAPK and NF-κB signaling inhibition in chondrocytes and synoviocytes on inflammatory condition. We sincerely appreciate your time, effort, and expertise in evaluating our manuscript. Your invaluable contributions have undoubtedly enhanced the quality of our work. We are grateful for the opportunity to benefit from your expertise and insights.

Thank you once again. We look forward to any further suggestions or guidance you may have as we continue to advance our research. In response to your valuable suggestions, we have made the following revisions:

Point 1. In Figures 1C and 1D, how much IL-1β was used to treat the cells? Were the cells treated with IL-1β + CA at concentrations ranging from 5 to 50 μM? If so, please re-label the figure, as the current labeling is very confusing.

Response 1: Thank you for your detailed feedback on our manuscript. In Figures 1C and 1D, the Chondrocytes and synoviocytes cells were treated with IL-1β at a concentration of 10 ng/ml for 24h  to assess the induction of inflammation. After that, There was treatment involving CA at concentrations ranging from 5 to 50 μM in these specific figures. This should help clarify the experimental conditions depicted in the figures. And as suggested the relabelling of the figure is being done.

Point 2.  Please revise the font and labeling in all figures. The current font and figure sizes are too small, making them difficult for readers to interpret.

Response 2: Thank you for pointing out the issues. As the figures labelling was uneven, it was changed to a proper size depicting all the figures in the manuscript.

Point 3. For assessing pro-inflammatory markers, why did the authors not include any ELISA data?

Response 3: We thank you for highlighting the ELISA, In our study, we chose to use Western blot analysis to depict the pro-inflammatory markers for several reasons:

Specificity and Sensitivity: Western blotting allows for the detection of specific protein bands corresponding to the pro-inflammatory markers with high specificity. This method enables us to verify the presence and relative expression levels of these proteins in a more detailed manner compared to ELISA.

Qualitative and Quantitative Analysis: Western blotting provides both qualitative and quantitative data. We were able to observe the presence of specific protein bands and also quantify the changes in protein expression using densitometry. This dual capability was crucial for our analysis, as it allowed us to detect subtle changes in protein expression that might not be as evident with ELISA alone

Point 4. Why does the illustration in Figure 5 focus on the NF-κB inhibitor, and the illustration in Figure 6 focus on U0126? The authors did not include any illustration describing how CA inhibits the pro-inflammatory signaling pathway in chondrocytes and synoviocytes in this manuscript.

Response 4: Thank you for suggesting the inclusion of CA role, Figure 5 focuses on the NF-κB inhibitor because NF-κB is a critical transcription factor in the regulation of pro-inflammatory responses. The illustration aims to depict how the NF-κB inhibitor (such as 5HPP-33) blocks the activation of NF-κB, thereby inhibiting the transcription of pro-inflammatory genes. This is a key mechanism through which the inhibitor exerts its anti-inflammatory effects. Figure 6 focuses on U0126, which is a MEK1/2 inhibitor, to illustrate its role in modulating the MAPK signaling pathway. This pathway is also involved in the regulation of inflammatory responses and cell proliferation. By inhibiting MEK1/2, U0126 prevents the phosphorylation of ERK, which in turn reduces the expression of pro-inflammatory genes and other downstream effects. In the revised figure, The specific mechanism by which CA inhibits pro-inflammatory signaling was included.

Point 5. In Figure 6, why did the authors use only the ERK inhibitor U0126? Based on the data in Figure 3, CA also effectively reduced IL-1β-induced p-JNK and p-p38 levels.

Response 5: Thank you for your insightful comments; In our study, we also investigated the NF-κB pathway, which is another critical signaling pathway involved in inflammation. By using U0126 for the ERK pathway and another inhibitor for the NF-κB pathway, we aimed to provide a balanced view of how different pathways contribute to the overall inflammatory response. There was no particular reason to exclude the other MAPK pathways (JNK and p38) entirely, but rather, we chose to focus on one representative pathway (ERK) to provide a detailed analysis from NF-kB and MAPK.

Point 6. Why did the authors use 10 μM of 5HPP-33 and U0126? Is there any data showing that this is the optimal dose?

Response 6: We appreciate your detailed feedback. Since our previous study also used 10 μM 5HPP-33 and U0126, we opted to maintain this concentration to ensure consistency and reliability of the data across different experiments.

https://pubs.acs.org/doi/10.1021/acsomega.4c01911

https://doi.org/10.3390/ijms241512282

Round 2

Reviewer 1 Report

Comments and Suggestions for Authors

Thank you.

Questions have been addressed.

Now manuscript is suitable for publication.

Author Response

Dear Reviewer,

We sincerely appreciate your time and effort in reviewing our manuscript and providing valuable feedback throughout this process. Your insightful comments and suggestions have significantly contributed to improving the quality and clarity of our work.

We are pleased to hear that our responses and revisions have adequately addressed your questions and concerns. We have made every effort to incorporate your suggestions and clarify the points you raised.

Thank you once again for your thorough evaluation and for deeming our manuscript suitable for publication. We appreciate your support and look forward to seeing our work published.
